# Synthesis of Nanocrystalline AZ91 Magnesium Alloy Dispersed with 15 vol.% Submicron SiC Particles by Mechanical Milling

**DOI:** 10.3390/ma12060901

**Published:** 2019-03-18

**Authors:** Shitian Su, Jixue Zhou, Shouqiu Tang, Huan Yu, Qian Su, Suqing Zhang

**Affiliations:** 1School of Mechanical and Electrical Engineering, Zaozhuang University, Zaozhuang 277160, China; susthit@163.com; 2Shandong Key Laboratory for High Strength Lightweight Metallic Materials, Advanced Materials Institute, Qilu University of Technology (Shandong Academy of Sciences), Jinan 250014, China; zhoujxhit@163.com (J.Z.); tangsqsdas@163.com (S.T.); zhangsqsdas@163.com (S.Z.); 3School of Materials Science and Engineering, Harbin Institute of Technology, Harbin 150001, China

**Keywords:** nanocrystalline, magnesium alloys, submicron SiC particles, hardness, mechanical milling

## Abstract

The development of a magnesium matrix composite with a high content of dispersions using conventional liquid-phase process is a great challenge, especially for nanometer/submicron particles. In this work, mechanical milling was employed to prepare nanocrystalline AZ91 dispersed with 15 vol.% submicron SiC particles (SiCp/AZ91). AZ91 with no SiCp was applied as a comparative study with the same mechanical milling. In order to investigate the mechanism of dispersing, the morphology evolution of powders and the corresponding SiCp distribution were observed. As the scanning electron microscope (SEM) analysis exhibited, the addition of SiCp accelerated the smashing of AZ91 particles, which promoted the dispersion of SiCp in AZ91. Thus, after mechanical milling, 15 vol.% SiCp, which was smashed from 800 to 255 nm, got uniformly distributed in the Mg matrix. Based on X-ray diffraction (XRD) results, part of the Mg_17_Al_12_ precipitate got dissolved, and an Al-supersaturated Mg solid solution was formed. The transmission electron microscopy (TEM) results showed that the ultimate Mg grain (32 nm) of milled SiCp/AZ91 was much smaller than that of milled AZ91 (64 nm), which can be attributed to a pinning effect of submicron SiCp. After mechanical milling, the hardness of SiCp/AZ91 reached 185 HV, which was 185% higher than the original AZ91 and 33% higher than milled AZ91, due to fine Mg grain and submicron dispersions.

## 1. Introduction

In recent years, due to the compelling need to achieve energy efficiency, safety, and human well-being, increasing attention is being paid to the development of novel metals that have high performance, are affordable, and have a lightweight structure [1,2,3]. Magnesium, which is abundant on Earth, is a light metal with low density (1.738 g/cm^3^) and ease of recycling. Thus, magnesium alloys have tremendous potential in fields of aerospace, railway, automobile, defense, electronics industries, and so on [4,5]. However, when applied to advanced structured materials, the advantages are eclipsed by the low strength of the element [6]. Therefore, it is highly meaningful to develop magnesium alloys with excellent mechanical properties.

Abundant studies [7,8,9] have confirmed that grain boundary strengthening is one of the most effective methods to achieve magnesium alloys with high strength. As is well known, materials with finer grains exhibit higher strength; this is especially the case with magnesium and its alloys with high Hall-Petch coefficient [10]. Hence, investigations on obtaining magnesium alloys with ultrafine grains have been carried out using various methods, including rapid solidification [11,12], severe plastic deformation [13,14,15], and mechanical alloying [16,17,18]. Sun et al. [19] prepared a nanostructured Mg-8.2Gd-3.8Y-1.0Zn-0.4Zr alloy using the method of high-pressure torsion (HPT). The size of the initial specimens was 10.0 mm in diameter and 1.0 mm in thickness. After HPT processing with 10 turns, magnesium nanocrystallization was achieved with an average grain size of approximately 48 nm. Wang et al. [16] prepared nanocrystalline (NC) Mg alloy powders by the processing of mechanically assisted hydriding and subsequent dehydriding. Massive Mg alloy powders were obtained with an average grain size of 25 nm. In general, considering the complexity of processing and the dimension of materials, mechanical alloying could be the most appropriate process to prepare nanocrystalline magnesium alloys. However, the strength of NC materials at elevated temperatures would decrease severely due to grain boundary slide [20,21]. Thus, it is particularly significant to improve the mechanical properties of NC magnesium alloys at elevated temperatures. Fortunately, the research on magnesium matrix composites reveals that particles, including ceramic and metal, would improve strength at various temperatures [8,22,23]. Meanwhile, the dispersions, as reinforced phase in the matrix, can pin up grain boundary, which is meaningful to keep high strength at room temperature for ultrafine grained (UG) magnesium alloys [7]. However, homogeneous dispersion, especially submicron/nanometer particles, in metal matrix is difficult to achieve [22,24]. Due to poor wettability between ceramic particles and the metal surface, segregation, which leads to coarse clusters being distributed along the matrix grain boundary, would always occur during metallurgy processing. Based on repeated welding, fracturing, and rewelding of powder particles, mechanical milling has been shown to have an outstanding ability to synthesize metal matrix composites with mass tiny dispersions [25].

Accordingly, nanocrystalline AZ91 magnesium alloy dispersed with 15 vol.% submicron SiC particles was obtained by mechanical milling. Meanwhile, the effect of SiC particles on the evolution of the microstructure, including powder morphology, phase transformation, and magnesium grain, was investigated by preparing an AZ91 Mg alloy (AZ91) with the same mechanical milling process. In addition, the strengthening mechanisms were analyzed quantitatively.

## 2. Experimental

### 2.1. Mechanical Milling Process

Mechanical milling was carried out using QM-DY4-type planetary ball mills. The weight ratio of milling balls (Ф 5 mm, Ф 8 mm, Ф 10 mm) to mixture powders was 60:1, and the weight of the mixture powders, including AZ91 chips and SiCp powders, was 10 g. To prevent agglomeration and excessive cold welding of powders, stearic acid was applied as the process control agent, and the weight ratio was 1%. The milling balls, mixture powders, and stearic acid were loaded into a stainless steel vial, and the vial was then sealed. All the above were performed in a glove box with an atmosphere of high-purity argon with standard atmospheric pressure. The milling process consisted of high-energy ball milling (400 rpm for 2 h) and low-energy ball milling (200 rpm for 2 h).

### 2.2. Microstructural Characterization

In this study, scanning electron microscope (SEM; Quanta 200FEG, Thermo Fisher Scientific, Hillsboro, OR, USA) was applied to observe the morphology of powders before and after mechanical milling. The microstructure of as cast AZ91 was observed by optical microscope (OM, Olympus GX71, Olympus, Tokyo, Japan). For the purpose of observing microstructure crossing powder particles, milled powders were compacted to green billets under a pressure of 2.0 GPa. Back-scattered electron (BSE) was used to observe the distribution of the second phase, including the Mg_17_Al_12_ phase and SiCp. Meanwhile, in order to detect the various phases, energy-dispersive spectrometer (EDS, Thermo Fisher Scientific, Hillsboro, OR, USA) was applied. To identify the microstructure, X-ray diffraction (XRD, Empyrean, Malvern Panalytical, Egham, UK) was performed with Cu Kα radiation. Transmission electron microscope (TEM, Talos f200x, Thermo Fisher Scientific, Hillsboro, OR, USA) was used to observe the microstructure of milled powders. The detailed description of the process to obtain TEM specimen is described in [18]. The hardness was tested by a microhardness tester (HV, HVS-1000Z, Shanghai Test Machinery Factory, Shanghai, China) with a load of 500 g and loading time of 15 s.

## 3. Results and Discussion

### 3.1. Characterization of Initial Materials

In this study, submicron SiC particles (SiCp) with average size of 800 nm, as shown in Figure 1a, were applied as reinforcements with the volume fraction of 15%. As-cast AZ91 with nominal composition of Mg-9.12Al-0.74Zn-0.25Mn was used as the base material. Figure 1b shows the morphology of AZ91 chips, which were ~80 μm in width and ~320 μm in length. The microstructure of the initial AZ91 is shown in Figure 1c–g. Based on the image of optical microscope (OM), the average grain size of magnesium was approximately 60 μm. The results of SEM and EDS showed that the initial AZ91 consisted of an Mg phase, an Mg_17_Al_12_ phase, and an Al_8_Mn_5_ phase. As shown in Figure 1d, the α-Mg phase was surrounded by the dendritic Mg_17_Al_12_ phase, and the Al_8_Mn_5_ phase appeared as spherical particles.

### 3.2. Morphology Evolution

During mechanical milling, the plastic deformation of the powders occurred due to the interactive collision between the powders, milling balls, and the pot. The microstructural evolution of AZ91 powders is described in Figure 2a–c. In the initial stage of high-speed mechanical milling, the powder particles got flattened, with the thickness decreasing and the width increasing. Meanwhile, the clustered β-phase, as reinforced phase in AZ91, began to be fragmented, as shown in Section 3.3. With continued mechanical milling, more dislocation pile-up was generated due to the repeat deformation. When the critical value of dislocation was reached, the powder particles were smashed. As shown in Figure 2c,d, the particles were smashed drastically, with the average particle size decreasing from 500 μm at 60 min to 60 μm at 120 min. A similar phenomenon was also observed in the high-speed mechanical milling of the AZ61 alloy [25]. After mechanical milling for 180 min, the AZ91 powders got refined further due to the addition of stearic acid. During mechanical milling, stearic acid adsorbs on the surface of the powder, which diminishes the possibility of cold welding between powder particles [26]. Therefore, the average particle size was reduced to approximately 30 μm, and agglomeration was inhibited. The average particle size was about 25 μm after mechanical milling for 240 min. The smashing of powder particles contributed to the dispersion of the β-phase, which is further analyzed in Section 3.3.

The morphology evolution of SiCp/AZ91 composite powders is shown in Figure 3. It has been found that the weight ratio of milling balls to powders plays a significant role in the time required to achieve a particular state in milled powders [25]. Due to the SiCp powders replacing the same quality AZ91 powders, the weight ratio of milling balls to AZ91 of SiCp/AZ91 composite powders increased to 73:1, which was higher than the value of AZ91 powders (60:1). Compared with the morphology evolution of AZ91 powders, the morphology of milled SiCp/AZ91 composite powders exhibited discrepant evolution. As shown in Figure 3a, all the composite powder particles were flattened after 15 min of mechanical milling, while a similar phenomenon for AZ91 powders appeared after 60 min. Moreover, most of the composite powder particles had been smashed, and the average particle size was about 300 μm. Therefore, it can be concluded that the composite powder particles were flattened and smashed more rapidly in mechanical milling processing. This may be attributed to the different weights of the initial AZ91 powders. As shown in Figure 5a, most of the SiCps distributed along the boundary of the AZ91 particles, thereby inhibiting the cold welding among powder particles. Thus, after mechanical milling for 30 min, the composite powders were evidently smashed, and the average particle size decreased to approximately 120 μm. However, due to the dispersion of SiCp in the AZ91 particles, as shown in Figure 5c, the hindering effect on cold welding, resulting from SiCp being aggregated along the particle boundaries, was weakened, and the particle size tended to be stable. During mechanical milling from 120 to 240 min, the composite powders were smashed further due to the addition of stearic acid. After mechanical milling, the average particle size of the composite powders was approximately 20 μm.

### 3.3. Dispersions Evolution

During mechanical milling, the powder particles got repeatedly welded, fractured, and rewelded, which allowed the area with the clustered β-phase to come in contact with the area without the β-phase. Accordingly, the β-phase gradually distributed uniformly in the magnesium matrix. The dispersion evolution of AZ91 is shown in Figure 4. After mechanical milling for 15 min, most of the β-phase was still in the segregation state, while some granular β-phase, labeled by white ellipse, was observed in the magnesium matrix. Due to repeated welding, fracturing, and rewelding of powder particles, the dispersion process of the β-phase accelerated. After mechanical milling for 120 min, it was found that only small amounts of β-phase, labeled by blue ellipses in Figure 4d, remained. With the smashing of powder particles due to the addition of stearic acid, the particle size of the dispersing β-phase decreased further. After mechanical milling for 180 min, there was still some big β-phase with particle size of about 10 μm. As shown in Figure 4f, the big β-phase particle vanished, and the β-phase dispersed in magnesium matrix reached submicron-scale with an average particle size of 700 nm.

Previous investigations [25,26,27] have found that, during mechanical milling, powder materials are in a nonequilibrium state; hence, supersaturated solid solutions may be generated by mechanical force. As shown in Figure 4, the amount of the β-phase decreased with the increase in mechanical milling time. Therefore, part of the β-phase decomposed, and the Al, which was rooted in the decomposed β-phase, dissolved into the magnesium matrix after mechanical milling.

During metallurgy processing, agglomeration of tiny particles will always distribute within grains and grain boundaries, especially to nanometer/submicron particles with high volume fraction [8,13]. The dispersion evolution of the SiCp/AZ91 composite, including the intrinsic β-phase and the added SiCp, is shown in Figure 5. Based on the EDS results of point A, B, and C, the dendritic area is β-phase, labeled by blue ellipses, while the granular area is the SiCp phase. After mechanical milling for 15 min, SiCps dispersed into only part of the AZ91 particles, and most of the SiCps distributed along the AZ91 particles. With the smashing of powder particles, as shown in Figure 3b, more SiCps entered into the AZ91 particles. The processing of repeated welding, fracturing, and rewelding resulted in most of the SiCps distributing uniformly in the AZ91 particles. However, there were still some areas, labeled by white ellipses, without SiCp, as shown in Figure 5d,e. Fortunately, with the addition of stearic acid, composite powders were evidently smashed, as shown in Figure 3e, which resulted in further dispersion of submicron SiCp. Thus, after mechanical milling for 180 min, all of the SiCps dispersed in the magnesium matrix, and no clustered SiCps were discovered. Meanwhile, some β-phase, labeled by blue ellipses, still remained, which was similar to the observation for AZ91 powders, as shown in Figure 4e. After mechanical milling, the added SiCps distributed uniformly, and the average particle size decreased from 800 to 255 nm. As shown in Figure 5g, the particle size distribution curve fitted well with the gauss theoretic curve. Although the size of some particles was about 1.2 μm, most of the SiCps were smaller than 300 nm. Therefore, it can be concluded that mechanical milling can achieve good dispersion of high volume fraction (15%) submicron SiCp in the magnesium matrix. As for particle-strengthened metal matrix composites, the strengthening phase should be distributed uniformly to achieve the desired strengthening effect. Based on the microstructural results for the dispersions, it can be confirmed that the mechanical milling processing used in this study is considerably useful in achieving Mg alloys dispersed with submicron/nanometer particles. Significantly, dispersions, acting as strengthening phase, got smashed further, which provided better contribution to the strength due to the Orowan strengthening mechanism. 

### 3.4. Phase Transformation

Figure 6 shows the XRD patterns of the initial AZ91, the milled AZ91, and the SiCp/AZ91 composite powders. As can be seen, the initial AZ91 was composed of the magnesium matrix and an Mg_17_Al_12_ phase. After mechanical milling, the Mg_17_Al_12_ phase was still present in milled AZ91, which is in accordance with the results of the microstructure shown in Figure 4f. In addition, the XRD pattern of milled SiCp/AZ91 composite powders showed that there were peaks of magnesium phase (No. 35-0821), Mg_17_Al_12_ phase (No. 01-1128), and SiCp phase (No. 49-1428). During mechanical milling, repeated plastic deformation of powders leads to dislocation pile-up. Therefore, the grain of the magnesium matrix was refined and the strain increased, which led to an increase in the full width at half maximum (FWHM). Taking the three most intensive peaks of magnesium matrix as an example, the peaks of magnesium matrix for milled AZ91 and SiCp/AZ91 were broadened in comparison to the initial AZ91, as shown in Figure 6. Based on XRD patterns and the previous calculation model [25], the values of FWHM for (100), (002), and (101) peaks are provided in Figure 7. The value of FWHM for AZ91 evidently increased after mechanical milling due to grain refining and microstrain accumulation. Meanwhile, the FWHM of milled SiCp/AZ91 was bigger than that of milled AZ91, which implied that the SiCp/AZ91 had finer grain or more microstrain. During mechanical milling, the movement of dislocations was hindered due to the pinning effect of dispersing SiCp, and the dislocation pile-up was thus accelerated. Therefore, the refining processing of magnesium grain was promoted, and more microstrain was formed.

During mechanical milling, powders are in a nonequilibrium state under the effect of high-energy collision [26]. In addition, with the refining of magnesium grain, the volume fraction of high-energy atoms belonging to grain boundaries increased significantly. Thus, solid solubility extension could be achieved, which affected the parameters of the magnesium matrix. Based on the results of the XRD patterns, the lattice parameters of the magnesium matrix for the initial AZ91, the milled AZ91, and the milled SiCp/AZ91 were calculated and are provided in Figure 8. As shown, both parameters *a* and *c* of the initial AZ91 were smaller than the theoretical values of pure Mg. Previous studies [25,28] have confirmed that the decrease in parameters can be attributed to the dissolution of Al in the Mg matrix. Meanwhile, compared with the initial AZ91, the parameters of milled AZ91 decreased further, which implied that the Al rooted in Mg_17_Al_12_ precipitates dissolved into the magnesium matrix during mechanical milling. In addition, the parameters of milled SiCp/AZ91 were smaller than the values of the initial AZ91, while they were larger than the values of milled AZ91. Thus, it can be concluded that, during mechanical milling, the SiCp hindered the dissolution of Mg_17_Al_12_ precipitates. As shown in Figure 5, after mechanical milling for 60 min, SiCp dispersed into most parts of the magnesium matrix. The distance between the magnesium atoms, near opposite side of SiCp, evidently increased, which weakened the diffusion of Al in the Mg matrix. Therefore, the dissolution of Mg_17_Al_12_ precipitates for SiC_p_/AZ91 decreased in comparison to AZ91.

### 3.5. Magnesium Matrix Analysis

The TEM images and the corresponding grain size distributions of milled AZ91 and SiCp/AZ91 are shown in Figure 9. After mechanical milling, the magnesium matrix of AZ91 with and without SiCp was nanostructured. High-energy collisions between the moving balls and the rotating milling pot resulted in plastic deformation of the powders. Thus the dislocation density increased gradually. As the dislocation density achieved a threshold value, the initial grain disintegrated to subgrains separated by small-angle grain boundaries. With the continuing plastic deformation, small-angle grain boundaries were displaced by large-angle grain boundaries. Thus, the grain of magnesium matrix was refined, and even NC was achieved. After mechanical milling, the average magnesium grain size of milled SiCp/AZ91 was approximately 32 nm. Meanwhile, the grain size distributed from 15 to 60 nm, the relative frequency of which corresponded with the Gauss fit. As shown in Figure 9c,d, the average grain size of milled AZ91 was about 64 nm, which was larger than that of milled SiCp/AZ91, and the grain size distribution was also consistent with the Gauss fit. The smaller average grain size of milled AZ91 with SiCp can be attributed to the submicron dispersions. During mechanical milling, the dispersions pinned up the movement of dislocation, which accelerated the dislocation pile-up. Therefore, the refining of the magnesium matrix was sped up, and finer magnesium grain was achieved. A similar phenomenon was also found and confirmed in previous studies [25,27].

### 3.6. Hardness Analysis

The microstructural evolution, including Mg grain refining/nanocrystallization, the dissolution of Mg_17_Al_12_ phase, and the dispersion of SiCp, affects the hardness to various degrees. Microhardness of the materials before and after mechanical milling is shown in Figure 10. After mechanical milling, the hardness of AZ91 increased from 65 to 137 HV, and the hardness of milled SiCp/AZ91 increased to 185 HV. Based on the relationship between the hardness and the yield strength (YS) [29], the strength of the initial AZ91, the milled AZ91, and the milled SiCp/AZ91 composite was ~217 MPa, ~457 MPa, and ~617 MPa, respectively. Thus, the increments of strength for milled AZ91 and SiCp/AZ91 composite were 240 MPa and 400 MPa, respectively. Previous studies have confirmed that the strengthening mechanisms in metal matrix composites include [8,30] (i) grain boundary strengthening ($\Delta \sigma_{\text{H-P}}$), (ii) solid solution strengthening ($\Delta \sigma_{SS}$), (iii) Orowan strengthening ($\Delta \sigma_{Orowan}$), and (iv) load-bearing strengthening ($\Delta \sigma_{Load}$).

As for the milled AZ91, the strengthening mechanisms were grain boundary strengthening and solid solution strengthening, which can be calculated using the following equations [31,32]:(1)$$\Delta \sigma_{\text{H-P}}=k_{\gamma }d^{-1/2}$$
(2)$$\Delta \sigma_{ss}=3^{3/2}\frac{G_{m}\delta^{3/2}c^{1/2}}{700}$$
where $k_{\gamma }$ is the Hall-Petch coefficient, d is the mean grain size, $G_{m}$ is the shear modulus of the matrix, ε is the mismatch parameter, and *c* is the atomic fraction of the solute. Previous studies [18,20,30] have confirmed that the strengthening effect originating in the dissolution of Al in Mg matrix is insignificant for UG magnesium alloys. A similar conclusion can also be arrived at by comparing milled AZ61 Mg and AZ91. As shown in the microstructural images in Figure 10b,c, the grain sizes of AZ61 Mg (~68 nm) and AZ91 (~64 nm) were very close. The lattice parameters of AZ91 were smaller than AZ61 with *a* being 0.31843 nm and *c* being 0.5174 nm [25], implying that more Al dissolved into the Mg matrix. Nevertheless, the hardness of milled AZ61 was almost the same as milled AZ91 [7]. Thus, in this study, based on the supposition that all Al dissolved into the magnesium matrix, the solid solution strengthening was calculated as ~6 MPa using Equation (2). Thus, the contribution of grain boundary strengthening to milled AZ91 was approximately ~234 MPa. Therefore, as with milled AZ91, the main strengthening mechanism was grain boundary strengthening, which accounted for 97%.

The strengthening contribution to the SiCp/AZ91 composite due to submicron SiCp by Orowan strengthening and the load-bearing mechanisms was calculated and analyzed further. With the same assumption about the dissolution of Al, the solid solution strengthening of the SiCp/AZ91 composite was about 6 MPa. The Orowan strengthening induced by well-dispersed submicron SiCp can be calculated by the following equation [8]:(3)$$\Delta \sigma_{Orowan}=\frac{\varphi G_{m}b}{d_{p}}{\left(\frac{6V_{p}}{\pi }\right)}^{1/3}$$
where b, $d_{p}$, and $V_{p}$ are the Burgers vector, the size, and volume fraction of SiCp, respectively. φ is a constant, equal to 2. Considering that, in this study, $G_{m}$ = 16.4 GPa, b = 0.32 nm, $V_{p}$= 0.15, and $d_{p}$ = 255 nm, the calculated $\Delta \sigma_{Orowan}$ was 27 MPa. 

The contribution of the load-bearing strengthening ($\Delta \sigma_{Load}$) can be calculated by the following equation [33]:(4)$$\Delta \sigma_{Load}=0.5V_{p}\sigma_{m}$$
where $V_{p}$ and $\sigma_{m}$ are the volume fraction of SiCp and the YS of the matrix, respectively. Meanwhile, the strength should be the magnesium matrix enhanced by grain boundary strengthening and solid solution strengthening. Thus, $\Delta \sigma_{Load}$ can be estimated to be 17 + 0.075$\Delta \sigma_{\text{H-P}}$ MPa. Overall, the $\Delta \sigma_{\text{H-P}}$ of SiCp/AZ91 composite was calculated to be 326 MPa, and the $\Delta \sigma_{Load}$ was then calculated to be 41 MPa. Therefore, the contribution ratio of grain boundary strengthening, solid solution strengthening, Orowan strengthening, and load-bearing strengthening to milled SiCp/AZ91 was about 81.5%, 1.5%, 6.8%, and 10.3%, respectively. It can be concluded that the addition of SiCp enhanced AZ91, not only by Orowan strengthening and load-bearing strengthening but also by better grain boundary strengthening. After the same mechanical milling, finer magnesium grain of SiCp/AZ91 composite was achieved due to submicron dispersions, which caused remarkable strength increment of 92 MPa.

A comparison of the hardness of various Mg alloys as well as the microstructural results, including particle size of dispersions and grain size of Mg, is shown in Table 1. The processing involved HPT, mechanical milling (MM), novel solidification processing method (NSP), casting, extrusion (ES), and thixoforging. As can be concluded by the analysis of hardness, the Mg grain size, dispersion size, and content were the main factors for strength. Due to the smallest Mg grain and submicron dispersion, the SiCp/AZ91 of this study possessed the highest hardness among the referred materials. Chen et al. [34] prepared Mg18Zn-6 vol.% SiCp by semisolid-state mechanical mixing with liquid-state ultrasonic processing. The dispersions with particle size of 50 nm were distributed uniformly in the Mg matrix, which led to high hardness of 183 HV. Therefore, it can be expected that the preparation of the composite, with more excellent mechanical properties using nanoscale SiCp, is available based on the same mechanical milling processing.

## 4. Conclusions

As an effective method to synthesize composites, mechanical milling was employed to achieve NC SiCp/AZ91. The effect of SiCp on the microstructure and mechanical property was investigated, and the following conclusions can be drawn.
(1)The addition of submicron SiCp accelerated the smashing of AZ91 particles during mechanical milling, which assisted in the dispersion of SiCp in the Mg matrix. After mechanical milling, the average particle size decreased from 800 to 255 nm, and the tiny SiCp, including some nanometer particles, was distributed uniformly in the Mg matrix.(2)During mechanical milling, powders are in a nonequilibrium state. The dissolution of Mg_17_Al_12_ precipitates was confirmed. Based on the lattice parameter evolution of the Mg matrix, Al-supersaturated Mg solid solution was formed. There were also some Mg_17_Al_12_ precipitates remaining, the pattern of which changed into particle with diameter of about 700 nm. Moreover, it was found that the SiCp dispersed in Mg matrix hindered the dissolution of Mg_17_Al_12_ precipitates.(3)After mechanical milling, NC AZ91 and SiCp/AZ91 were achieved. Meanwhile, due to the pinning effect on the movement of dislocations, the dislocation pile-up of SiCp/AZ91 was accelerated. Thus, the average grain size of milled SiCp/AZ91 (~32 nm) was smaller than that of milled AZ91 (~64 nm).(4)After mechanical milling, the hardness of SiCp/AZ91 was approximately 185 HV, which was 185% higher than that of the initial AZ91. The strengthening mechanisms of SiCp/AZ91 were analyzed quantitatively. The contribution ratio of grain boundary strengthening, solid solution strengthening, Orowan strengthening, and load-bearing strengthening to milled SiCp/AZ91 were about 81.5%, 1.5%, 6.8%, and 10.3%, respectively. Meanwhile, due to the addition of submicron SiCp, the hardness of SiCp/AZ91 increased by 33% compared with milled AZ91 due to Orowan strengthening, load-bearing strengthening, and more enhanced grain boundary strengthening.

## Figures and Tables

**Figure 1 materials-12-00901-f001:**
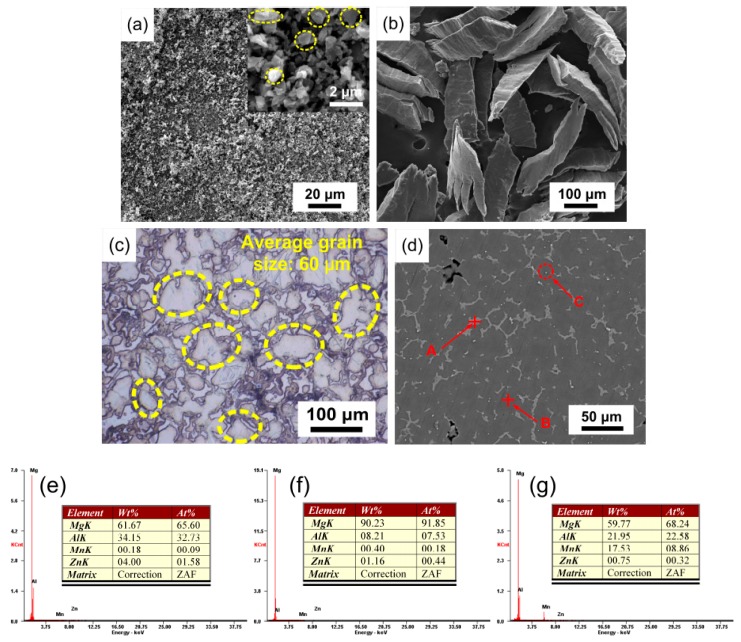
Microstructure of initial materials: (**a**,**b**) morphology of SiCp powders and AZ91 chips; (**c**,**d**) optical microscope (OM) and scanning electron microscope (SEM) images of AZ91; (**e**–**g**) energy-dispersive spectrometer (EDS) results of points A, B, and C in Figure 1d.

**Figure 2 materials-12-00901-f002:**
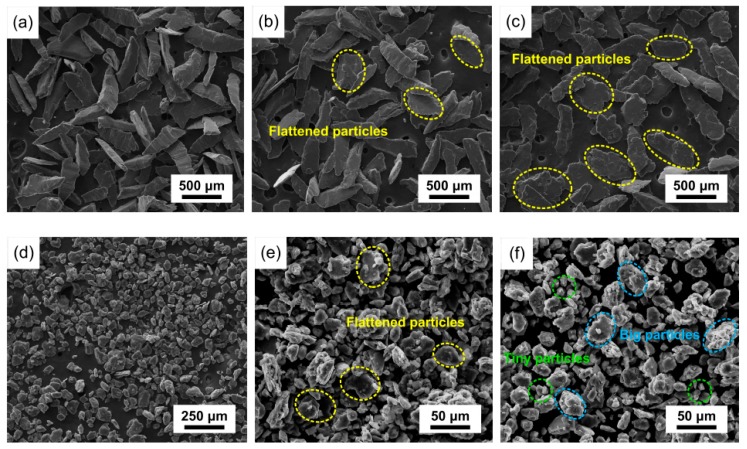
Morphology evolution of AZ91: (**a**) 15 min; (**b**) 30 min; (**c**) 60 min; (**d**) 120 min; (**e**) 180 min; (**f**) 240 min.

**Figure 3 materials-12-00901-f003:**
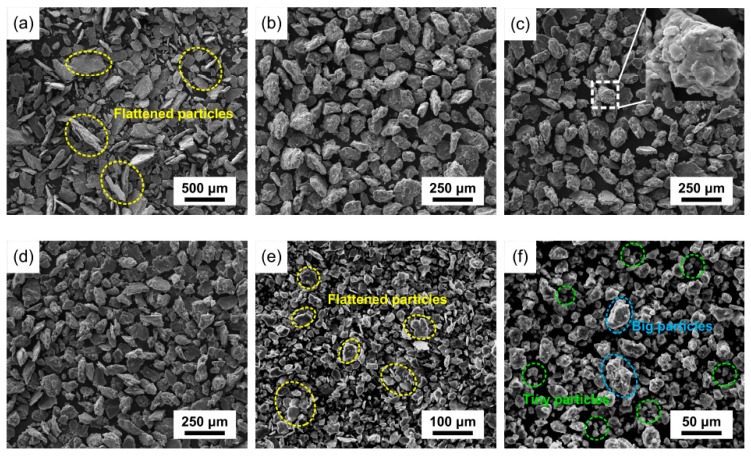
Morphology evolution of SiCp/AZ91 composite: (**a**) 15 min; (**b**) 30 min; (**c**) 60 min; (**d**) 120 min; (**e**) 180 min; (**f**) 240 min.

**Figure 4 materials-12-00901-f004:**
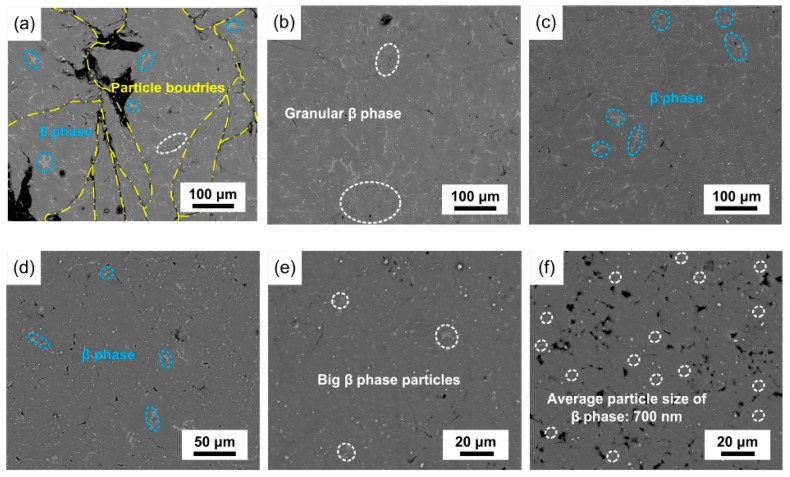
Microstructural evolution of AZ91: (**a**) 15 min; (**b**) 30 min; (**c**) 60 min; (**d**) 120 min; (**e**) 180 min; (**f**) 240 min.

**Figure 5 materials-12-00901-f005:**
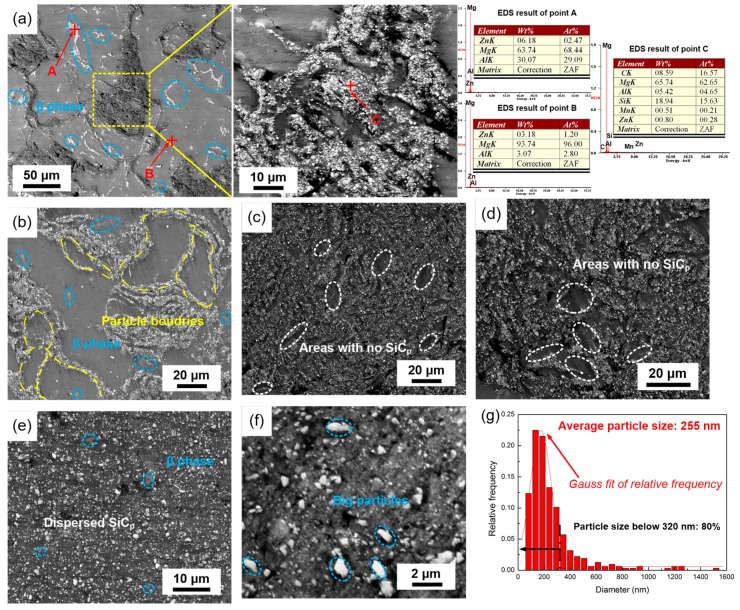
Microstructural evolution of SiCp/AZ91 composite: (**a**) 15 min; (**b**) 30 min; (**c**) 60 min; (**d**) 120 min; (**e**) 180 min; (**f**) 240 min (**g**) the particle size distribution of Figure 5f.

**Figure 6 materials-12-00901-f006:**
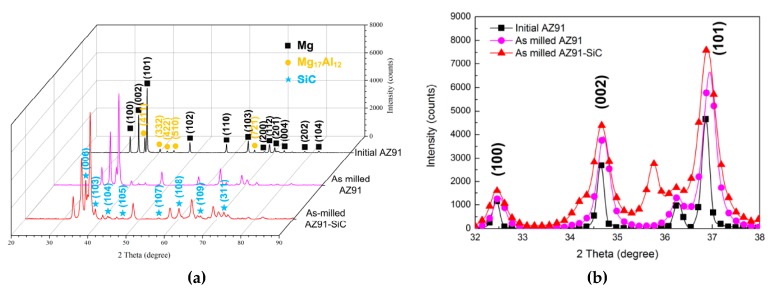
X-ray diffraction (XRD) patterns of the initial AZ91, the milled AZ91, and the SiCp/AZ91 composite powders. (**a**) Three-dimensional XRD patterns; (**b**) Two-dimensional XRD patterns.

**Figure 7 materials-12-00901-f007:**
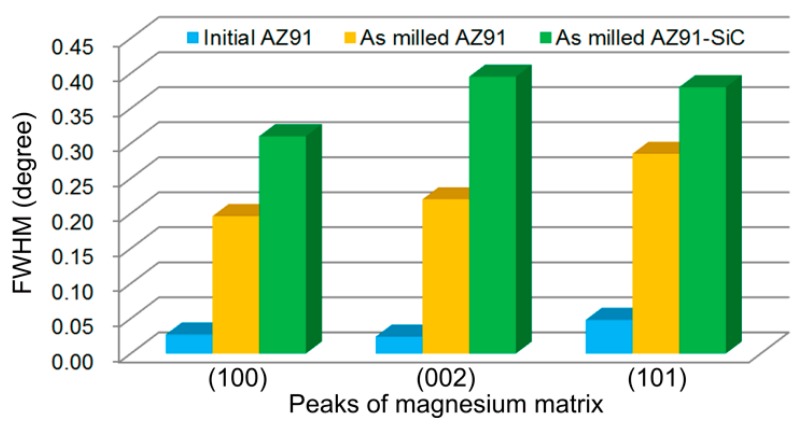
The full width at half maximum (FWHM) values of the three most intensive peaks of the magnesium matrix.

**Figure 8 materials-12-00901-f008:**
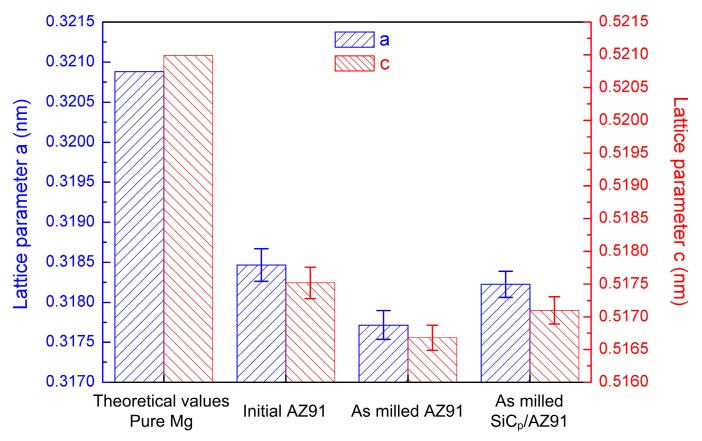
The lattice parameters of magnesium matrix with various states.

**Figure 9 materials-12-00901-f009:**
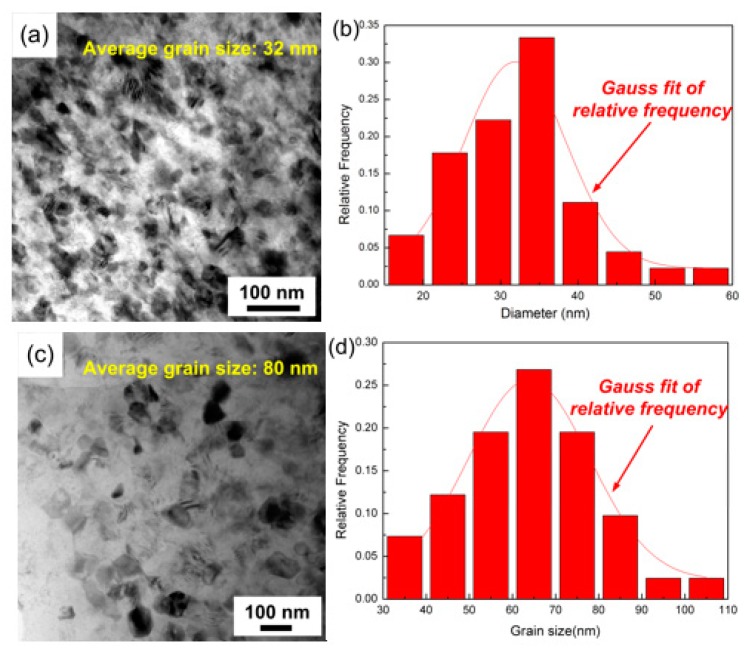
Transmission electron microscope (TEM) images and corresponding grain size distribution of milled SiCp/AZ91 (**a**,**b**) composite and AZ91 (**c**,**d**) powders.

**Figure 10 materials-12-00901-f010:**
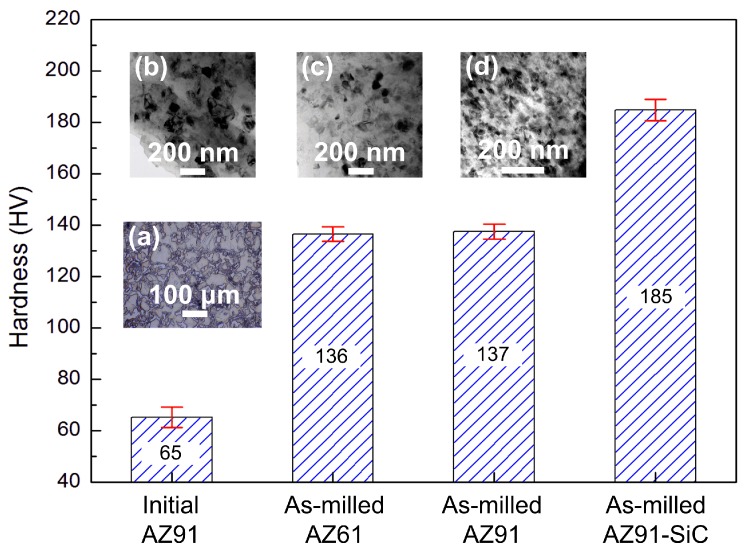
Hardness magnesium alloys with various states. (**a**) initial AZ91; (**b**) as-milled AZ61; (**c**) as-milled AZ91; (**d**) as-milled AZ91-SiC.

**Table 1 materials-12-00901-t001:** Comparison of microstructure and hardness of various Mg alloys.

Materials	Particle Size (μm)	Grain Size (μm)	Hardness (HV)	Processing
AZ31B	-	0.12	123	HPT [35]
Mg-8.2Gd-3.8Y-1.0Zn-0.4Zr	-	0.10	145	HPT and annealing [19]
AZ61-8 vol.% Ti	0.36	0.05	147	MM [7]
Mg-1.11 vol.% Al_2_O_3_	<0.1	31	70	MM and sintering [36]
Mg-2.5 wt.% TiB_2_	0.1–0.7	-	107	As-cast [37]
Mg18Zn-6 vol.% SiCp	0.05	-	183	NSP [34]
Mg-10 vol.% SiCp	0.05	0.06	141	MM [38]
Mg-10 vol.% SiCp	0.05	0.16	99	MM and ES [38]
AZ91-15 vol.% SiCp*	10	48	133	Thixoforging [39]
AZ91-15 vol.% SiCp	0.26	0.03	185	MM

* The volume fraction of SiCp of measurement point is about 50%. HPT, high-pressure torsion; MM, mechanical milling; NSP, novel solidification processing method; ES, extrusion.

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
