# Peer review of "Synthesis of Nanocrystalline AZ91 Magnesium Alloy Dispersed with 15 vol.% Submicron SiC Particles by Mechanical Milling"

_materials, 2019, doi:10.3390/ma12060901_

Round 1
Reviewer 1 Report
The submitted work concerns the development of nanotechnology for obtaining New nanomaterials. It is a very interesting study that adds new data to synthesis of nanocrystalline AZ91 magnesium alloy dispersed with submicron SiC particles by mechanical milling. The topic and scope of the manuscript is highly suitable for the readership of the Materials. This work is characterized by a high scientific level and a high degree of originality of the results and is recommended for publication in the Materials after minor corrections.
Some detailed comments are as following:
1). The subject matter is appropriate for the Materials.
2). The quality of the presentation is high.
3). The work contains new and original contributions.
4). Any lack of clarity.
5). Any theoretical errors.
6). Appropriate reference to previous work is given.
7). The conclusions are sound and justified.
8). The abstract is informative.
9). There is no material which might be omitted.
10). The main comment is not to present the experimental results of the material in the initial state in point 2. Experimental, 2.1. Materials. The experimental part should include a description of sample preparation, methods used, apparatus and test conditions. That is why Figure 1 and its description should be moved to part 3. Results and discussion. Please, correct.
11). In Results and discussion, the Authors have to add the numbers of ICDD PDF card used for the phase identification of the tested materials.
Author Response
Response to Reviewer 1 Comments
Point 1: The subject matter is appropriate for the Materials.
Response 1: Thanks for the reviewer’s positive comments.
Point 2:The quality of the presentation is high.
Response 2:Thanks for the reviewer’s positive comments.
Point 3: The work contains new and original contributions.
Response 3: Thanks for the reviewer’s positive comments.
Point 4: Any lack of clarity.
Response 4: Thanks for the reviewer’s positive comments.
Point 5: Any theoretical errors.
Response 5: Thanks for the reviewer’s positive comments.
Point 6: Appropriate reference to previous work is given.
Response 6: Thanks for the reviewer’s positive comments.
Point 7: The conclusions are sound and justified.
Response 7: Thanks for the reviewer’s positive comments.
Point 8: The abstract is informative.
Response 8: Thanks for the reviewer’s positive comments.
Point 9: There is no material which might be omitted.
Response 9: Thanks for the reviewer’s positive comments.
Point 10:The main comment is not to present the experimental results of the material in the initial state in point 2. Experimental, 2.1. Materials. The experimental part should include a description of sample preparation, methods used, apparatus and test conditions. That is why Figure 1 and its description should be moved to part 3. Results and discussion. Please, correct.
Response 10: Thanks for the reviewer’s comments. We totally agree with the suggestion that the experimental part should include a description of sample preparation, methods used, apparatus and test conditions. And the corresponding description about initial materials has been moved to part 3. Results and discussion.
Point 11: In Results and discussion, the Authors have to add the numbers of ICDD PDF card used for the phase identification of the tested materials.
Response 11: Thanks for the reviewer’s comments. According to the suggestion, the numbers of ICDD PDF card used for the phase identification of the tested materials have been added in the revised manuscript.

Reviewer 2 Report
The authors investigated the morphology, microstructural characterization, and mechanical hardnesss of SiC/AZ91. The hardness of SiCp/AZ91 reached to 185 HV, being 185% higher than original AZ91 and 33% higher than milled AZ91. Those results are very interesting.
1. L5 Authors names : write all authors names in full.
2. L217-219 : "During mechanical milling, the movement of dislocations is hindered due to the pinning effect of dispersing SiCp and thus the dislocation pile up is accelerated. Therefore, the refining processing of magnesium grain is promoted and more microstrain is formed. "
Is there any other experimental examples or results?
3. How is the ductility of the samples AZ91, and AZ91-SiC?
Author Response
Point 1: L5 Authors names : write all authors names in full.
Response 1: Thanks for the reviewer’s comments. According to the suggestion, all authors names in full have been written in the revised manuscript.
Point 2: L217-219 : "During mechanical milling, the movement of dislocations is hindered due to the pinning effect of dispersing SiCp and thus the dislocation pile up is accelerated. Therefore, the refining processing of magnesium grain is promoted and more microstrain is formed. " Is there any other experimental examples or results?
Response 2: Thanks for the reviewer’s comments. According to the suggestion that other experimental examples or results should be provided, the description that “Similar phenomenon was also found and confirmed in the previous studies [25, 27]” was added in the revised version of the manuscript.
Point 3: How is the ductility of the samples AZ91, and AZ91-SiC?
Response 3: Thanks for the reviewer’s comments. We totally agree with the suggestion that ductility is another significant mechanical property, study on the ductility of the samples AZ91, and AZ91-SiC would clarify the effect the nanocrystalline Mg matrix and submicron SiC particles on plasticity. According to reviewer’s suggestion, we will analyze the mechanical properties further, including ductility and strength, after the preparation of AZ91 and AZ91-SiC bulks by the milled powders in the following investigation.
